# A Bayesian approach to modeling phytoplankton population dynamics from size distribution time series

Jann Paul Mattern[1], Kristof Glauninger[2,3], Gregory L. Britten[4], John R. Casey[4,5], Sangwon Hyun[6], Zhen Wu[4], E. Virginia Armbrust[2], Zaid Harchaoui[3], François Ribalet[2]*

**1** Ocean Sciences Department, UC Santa Cruz, Santa Cruz, California, United States of America, **2** School of Oceanography, University of Washington, Seattle, Washington, United States of America, **3** Department of Statistics, University of Washington, Seattle, Washington, United States of America, **4** Program in Atmospheres, Oceans, and Climate, Massachusetts Institute of Technology, Cambridge, Massachusetts, United States of America, **5** Department of Oceanography, University of Hawai'i at Manoa, Honolulu, Hawaii, United States of America, **6** Department of Data Sciences and Operations, University of Southern California, Los Angeles, California, United States of America

☯ These authors contributed equally to this work.
* ribalet@uw.edu

**Data Availability Statement:** We value scientific reproducibility and have made all data and software code used to generate results and figures available

## Abstract

The rates of cell growth, division, and carbon loss of microbial populations are key parameters for understanding how organisms interact with their environment and how they contribute to the carbon cycle. However, the invasive nature of current analytical methods has hindered efforts to reliably quantify these parameters. In recent years, size-structured matrix population models (MPMs) have gained popularity for estimating division rates of microbial populations by mechanistically describing changes in microbial cell size distributions over time. Motivated by the mechanistic structure of these models, we employ a Bayesian approach to extend size-structured MPMs to capture additional biological processes describing the dynamics of a marine phytoplankton population over the day-night cycle. Our Bayesian framework is able to take prior scientific knowledge into account and generate biologically interpretable results. Using data from an exponentially growing laboratory culture of the cyanobacterium *Prochlorococcus*, we isolate respiratory and exudative carbon losses as critical parameters for the modeling of their population dynamics. The results suggest that this modeling framework can provide deeper insights into microbial population dynamics provided by size distribution time-series data.

## Author summary

Inferring the growth and population dynamics of marine microorganisms in their natural habitat is crucial to understanding the flow of carbon in the natural environment but remains a grand challenge due to the invasive nature of current measurement methods. As time-series observations of microorganism size distributions have become more commonplace in aquatic environments, matrix population models (MPMs), which aim to

at https://github.com/CBIOMES/bayesian-matrix-population-model.

**Funding:** This work was supported by grants from the Simons Foundation (no. 549945 to E.V.A, no. 574495 to F.R., no. 549894 to J.C.) and the Institute for Foundations of Data Science (IFDS; grant no. TRIPODS DMS 2023166 to Z.H.). G.L.B was supported by the Simons Foundation Postdoctoral Fellowship (no. 645921) in Marine Microbial Ecology. S.H., J.P.M, and Z.W. acknowledge the support of research funds from the Simons Foundation. The funders had no role in study design, data collection and analysis, decision to publish, or preparation of the manuscript.

**Competing interests:** The authors have declared that no competing interests exist.

mechanistically describe the change in size distribution over time, have gained in popularity over the last decade to estimate rates of cell division of these populations. Here, we build upon this work to improve accuracy and interpretability of model output and assess the relevance of previously omitted biological processes. We evaluated the performance of our models on a dataset of laboratory experiment time-series measurements of the cyanobacterium *Prochlorococcus*, Earth's most abundant photosynthetic organism, and demonstrated improved accuracy of division rate estimates by incorporating respiratory and exudative carbon losses into the modeling of their population dynamics.

## Introduction

Marine phytoplankton are photosynthetic microorganisms that account for up to half of global net primary production [1]. As such, the population dynamics of these organisms are crucial to understanding the global carbon cycle [2, 3]. One key aspect of phytoplankton populations is the growth rate, typically defined as the rate of increase in population biomass over time per unit of existing biomass. Direct *in-situ* measurement of this quantity cannot be obtained from abundance or carbon biomass alone, which are a composite of cell growth, cell mortality, and other biological and physical processes [4]. Several different methodologies have been employed to estimate *in-situ* phytoplankton growth rates; however, previous estimates relied on analytically challenging and low-throughput methods such as the radiometric turnover times of $^{14}$C labeled chlorophyll [5] and $^{32}$P labeled ATP [6], cell cycle analysis [7], and the dilution method [8]. While taxon-specific growth rates might be estimated with these methods, they often suffer from large uncertainties caused by coarse sample time resolution or experimental artifacts (collectively known as "bottle effects"; e.g., [9]). The emergence of continuous flow cytometry in ocean surveys [10–12] provides high resolution, taxon-specific measurements of the abundance and size of individual phytoplankton cells and offers a high-throughput *in-situ* alternative. In principle, measurements of cell abundance across different sizes over time provide a means to derive rates of carbon fixation and cell division [4], motivating the use of size-based mechanistic modeling frameworks to isolate these biological rates.

We focus on a class of mechanistic models known as stage-structured matrix population models (MPMs), which can be used to estimate demographic rates from measurements of abundance across life-cycle stages [13], often defined by the age or size of individuals. For example, tree species produce seeds once they have reached a particular size [14] and fish species maximize reproduction at a critical age [15]. These models assume that individuals in a population can be classified into *m* discrete stages that define their response to the environment modeled as a discrete-time process. MPMs assume that the state of the population at time $t + 1$ can be written in terms of the state of the population at time *t* and a set of transition rates [16]:

$$\boldsymbol{n}_{t+1} = \boldsymbol{B}_t(\boldsymbol{\theta})\, \boldsymbol{n}_t, \tag{1}$$

where $\boldsymbol{B}_t(\boldsymbol{\theta})$ is a *projection matrix* that defines the possibly time-dependent population dynamics, $\boldsymbol{\theta}$ is a parameter vector, and $\boldsymbol{n}_t$ is a vector representing the number of individuals in each stage at time *t*, which defines the state of the population. The vector $\boldsymbol{\theta}$ includes biological parameters to model population dynamics and is the target of parameter estimation [17].

In recent years, size-structured MPMs have gained popularity for estimating division rates of phytoplankton populations by mechanistically describing changes in microbial cell size distributions over the day-night cycle [18–24]. For instance, MPMs have been employed to

estimate daily division rates of the picocyanobacterium *Synechococcus* and picoeukaryotic phytoplankton based on a 13-year hourly time series from a coastal location in the Atlantic Ocean using a submersible flow cytometer [19, 23, 24]. In the North Pacific Subtropical Gyre, similar MPMs were used to estimate daily and hourly division rates of another picocyanobacterium, *Prochlorococcus*, based on continuous flow cytometry measurements taken over two research cruises [21]. In these studies, cell size measurements provided by high-frequency flow cytometry were used to define the life-cycle stages of the population. These models assumed that changes in the cell size distribution over the day-night cycle are driven by two biological processes: 1) carbon fixation via photosynthesis and 2) cell division; other processes such as respiration and exudation, which lead to cell shrinkage, are omitted. In previous investigations, model results were validated against estimates from dilution experiments [19] or DNA-based cell cycle analysis [21]. The focus of these models was to estimate division rates rather than carbon fluxes; as a consequence, these MPMs [18, 19, 21, 24] can lead to transition matrices with biologically implausible estimates. Furthermore, uncertainty quantification for model parameters typically involved refitting methods.

Here, we extend existing size-structured MPMs to model additional biological processes describing population dynamics over the day-night cycle and to improve parameter interpretability and model performance. Model estimates are computed using a Bayesian implementation in the probabilistic programming language Stan [25], through which we provide statistically rigorous parameter uncertainty intervals while allowing for the incorporation of prior scientific knowledge about model parameters. This approach enabled an evaluation of the sensitivity of posterior distributions to sampling size, sampling frequency, and initial conditions. In the following, we develop five MPMs that differ in their complexity and flexibility in parameterizing three transition terms: cell division, carbon fixation, and carbon loss (Fig 1), which describe the dynamics of the picocyanobacterium

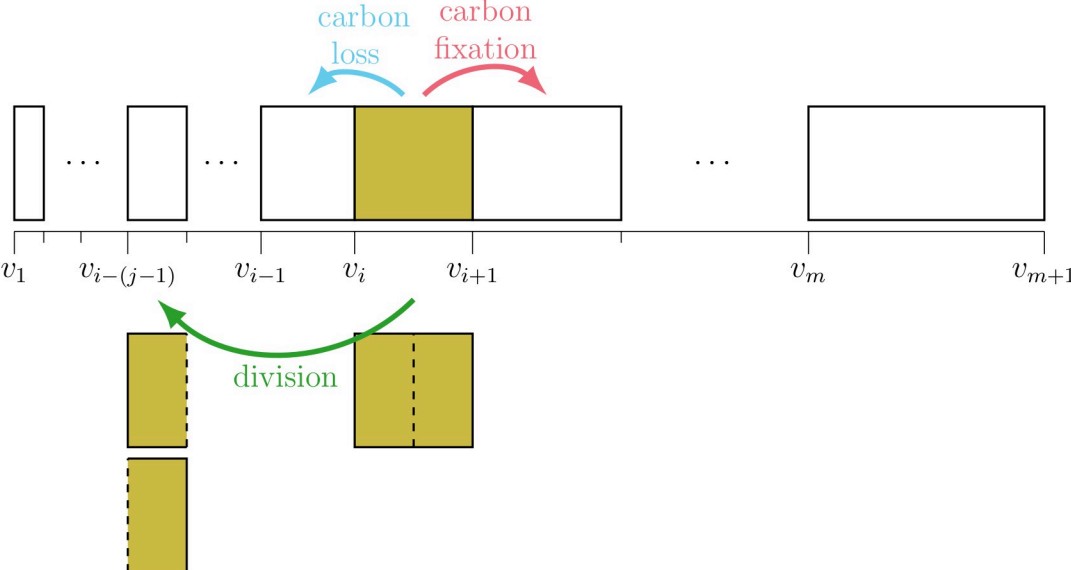

**Fig 1. MPM size classes and transitions.** Schematic of the MPM's cell size classes and its three class transitions: carbon fixation, division, and carbon loss. The boundaries of the $m$ cell size classes ($v_i$ for $i = 1, 2, \ldots, m + 1$) are logarithmically spaced, so that cells can transition to a size class that is exactly half their original size when they divide. For this purpose, the integer $j$ is selected so that $v_{i-(j-1)} = \frac{1}{2} v_i$ for $i \geq j$; cells in the first $j-1$ size classes cannot divide.

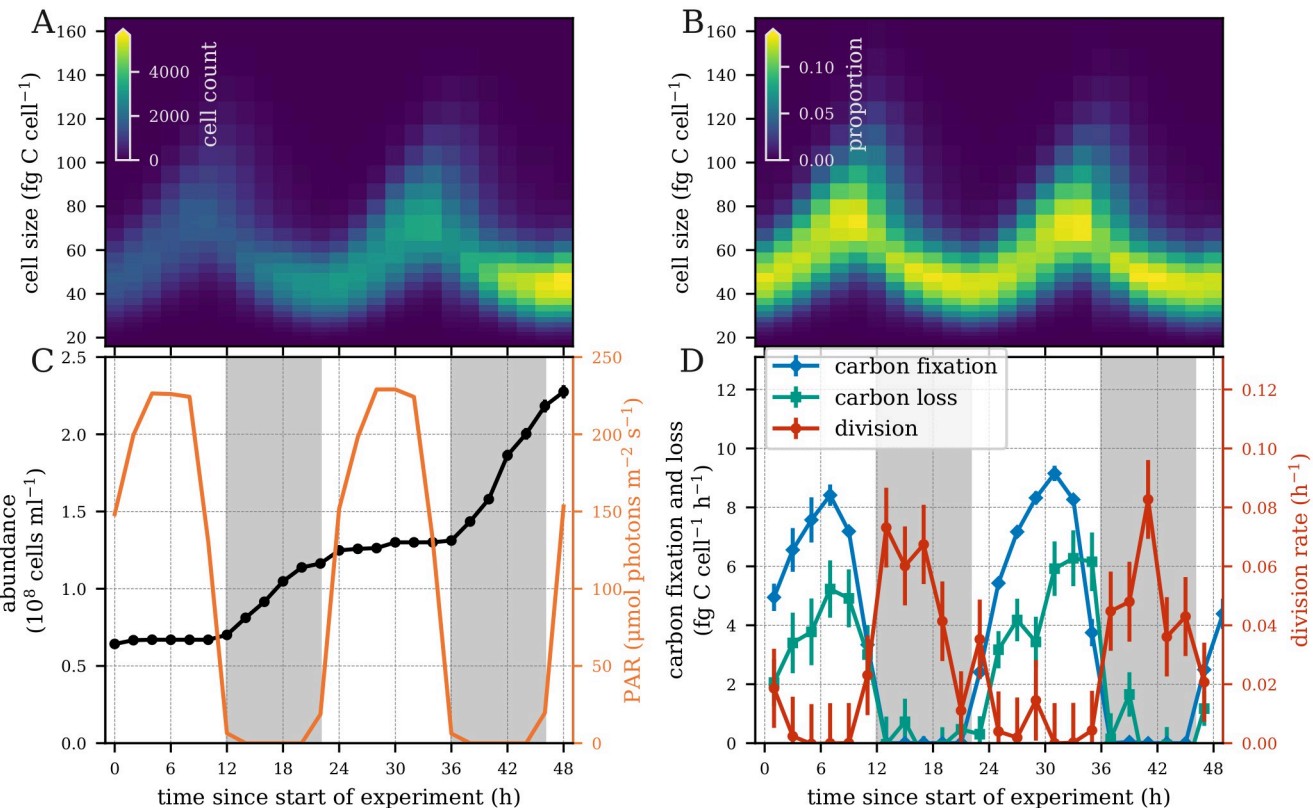

**Fig 2. Laboratory *Prochlorococcus* time series measurements.** (A) Heatmap of the number of cells and (B) relative cell abundances in each size class measured every two hours over a 48-hour period. (C) Cell abundance and photosynthetically active radiation (PAR). (D) Hourly carbon fixation, carbon loss, and division rates. Error bars indicate one standard deviation based on two technical and two biological replicates.

*Prochlorococcus*, Earth's most abundant phytoplankton [26]. Four additional models (S1 Table) with intermediate levels of complexity, are examined in the Supporting Information (S1 and S2 Figs).

We evaluated the performance of our models using laboratory experiment time-series measurements of a highly synchronized population of a high-light adapted strain of *Prochlorococcus* [27] collected during the exponential phase of batch growth over two simulated day-night cycles (Fig 2). This dataset contains cell size distributions derived from flow cytometry (Fig 2A and 2B), cell abundance and light measurements (Fig 2C), and measurements of carbon fixation (Fig 2D) at two-hour intervals. Division rates are derived from changes in cell abundances, while carbon loss is estimated from other measurements (see Experimental data below). We fit our models to the size distribution data (Fig 2A and 2B) and evaluated the ability of each model to reproduce the observed parameters at daily and hourly time scales. All models used a logarithmically-spaced discrete cell size distribution, permitting cells to divide into two daughter cells that are half their size (Fig 1). While our simplest model has no size-dependence for carbon fixation and lacks a carbon loss term, the more complex models include size-dependence for all three transitions, explained below. Finally, we converted model parameters to estimates of biological rates such as carbon fixation and carbon loss, allowing for more direct interpretation of estimated parameter values.

## Results

### Models

Past work has assumed that changes in cell size result from two processes: carbon fixation and cell division [18–24]. We built upon these studies by developing models that include an additional process: cell shrinkage through exudation and respiration. Another assumption of past models is that division is a monotonically increasing function of size, i.e. larger cells are more likely to divide than smaller cells. This implies that the highest rates of cell division should coincide with the highest proportion of large cells in the size distribution. However, the peak of cell size in *Synechococcus* and *Prochlorococcus* occurs during daylight while the peak of division usually occurs at night [28]. In the *Prochlorococcus* culture dataset used in our work, hourly cell division lagged 4–8 hours behind the peak of cell size (Fig 3A). In fact, hourly division rates showed little correlation with mean cell size (Fig 3B). Note that these trends hold not only for the mean cell size, but also with respect to larger cells, e.g. for the 70th, 80th, 90th, and 95th percentiles. When comparing the size distribution at 15 hours (peak in cell division) and at 35 hours (almost no division) after the start of the experiment, we see that many more large cells are present at hour 35, but the division rate is much higher at hour 15 (Fig 3C). However, we observed a strong correlation (r = 0.84) between hourly division rate and mean cell size with a 6-hour lag (Fig 3D). These results were also consistent for the 70th, 80th, 90th, and 95th percentiles, suggesting that cell division is dependent on cell size as well as additional

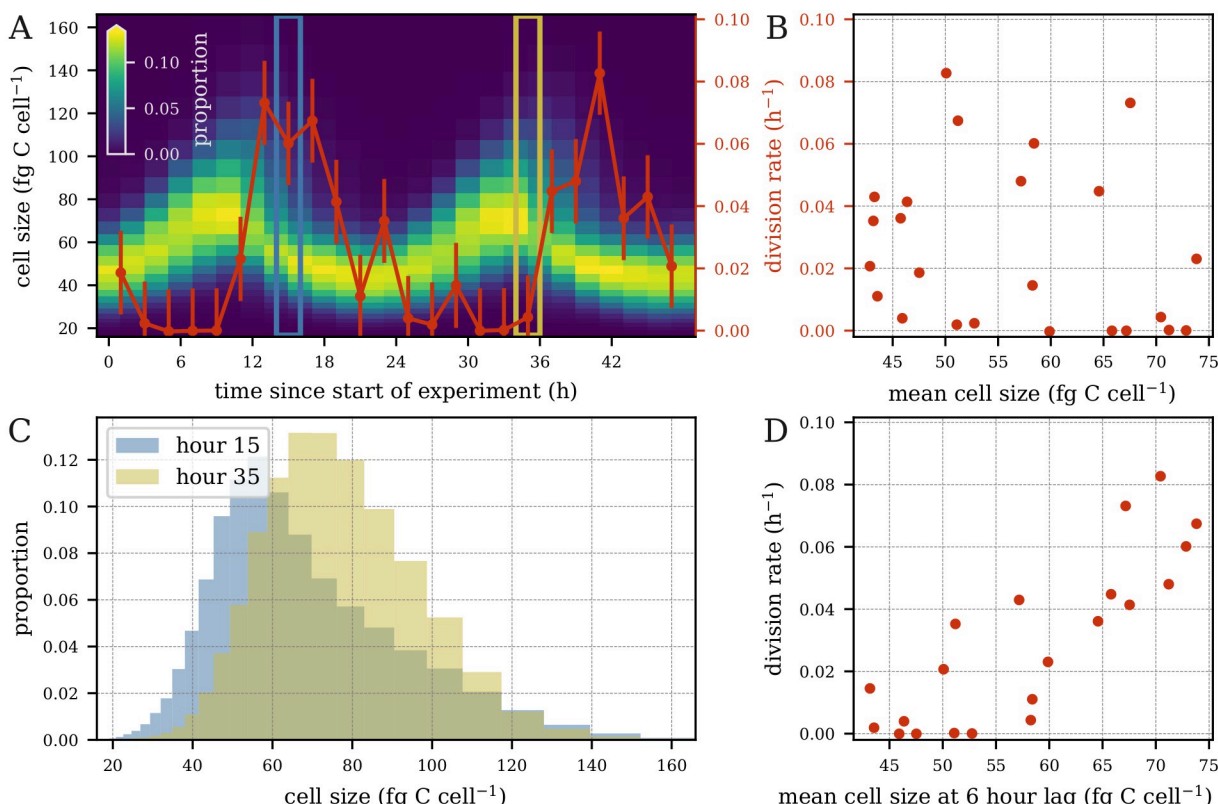

**Fig 3. Hourly division rates vs. cell size.** (A) Phytoplankton size distribution overlaid with hourly division rates (red curve; error bars indicate one standard deviation based on two technical and two biological replicates). Division rate and size distribution at *t* = 15 (blue box) and *t* = 35 (gold box). (B) Hourly division rates vs. mean cell size. (C) Cell size distribution at time *t* = 15 (blue) and *t* = 35 (gold). (D) hourly division rate at time *t* vs. mean cell size at time *t* − 6.

processes. Cell division in photosynthetic organisms is tightly regulated by light, although the onset of the cell cycle in *Prochlorococcus* does not seem to be strictly light dependent [29]. We therefore tested two different parameterizations for estimating cell division. In the first, cell division is constrained to be an increasing monotonic function of cell size, but constant over time, as in previous studies. In the second, cell division still increases monotonically with cell size but is allowed to vary over time. We also considered size dependence in carbon fixation through power-law relationships supported by experimental evidence [30]. Finally, we implemented a "free" parameterization in which carbon fixation and carbon loss rates are estimated separately for each size class, in order to provide enough flexibility for the model to capture biological processes that are not explicitly accounted for in our models.

We distilled our assumptions into a set of five models of differing parameterizations (Table 1). Each model is identified by a subscript consisting of three letters corresponding to the parameterizations of carbon fixation, division, and carbon loss, respectively. The first letter in each model name corresponds to the carbon fixation parameterization. The letter **b** in carbon fixation indicates a basic parameterization in which carbon fixation is assumed to be constant as a function of size. The letter **p** indicates a power-law relationship with respect to size and **f** represents a free parameterization where each size class may have its own rate of carbon fixation. With respect to division, represented by the second letter of the model name, the letter **m** indicates a monotone increasing division rate as a function of size with no time-dependence, while **t** indicates a parameterization that also includes time-dependence in division. The third letter, indicating the carbon loss parameterization, can be **b** (basic) or **f** (free parameterization) as in carbon fixation, or **x** for a model with no carbon loss. As an example, we refer to our simplest model as $m_{bmx}$, denoting that it has basic carbon fixation without size-dependence, division rates that monotonically increase with cell size, and no carbon loss term. Examples of the functional forms used for the model parameters can be found in the Materials and methods section.

Models contain more parameters down the rows of Table 1. Thus, model $m_{bmx}$ is the simplest model and most closely represents previous MPMs applied to microbial communities, while model $m_{ftf}$ is the most complex with respect to the number of parameters. We fit $m_{bmb}$ and $m_{ftf}$ on model-generated data to verify that our models were able to recover the values of the biological rate parameters (S3 Text).

We fit these five models to a dataset gathered in a laboratory experiment. Rates of division, carbon fixation, and carbon loss were estimated on both daily and hourly timescales. In the following section, we examine daily rate estimates, which have been the primary target of inference in past work. Then, we further assess the model rate estimates at an hourly timescale to inspect the behavior of our models within diel cycles. Furthermore, we explore the relationship between cell size and division, carbon fixation, and carbon loss. Finally, we examine the relationships between the estimated parameter values and perform observation sensitivity experiments.

**Table 1. Key models.**

| Model[*] | Growth | Division | Loss |
|---|---|---|---|
| $m_{bmx}$ | **b**asic | **m**onotonic | **x** (no loss) |
| $m_{bmb}$ | **b**asic | **m**onotonic | **b**asic |
| $m_{pmb}$ | **p**ower-law size-dependence | **m**onotonic | **b**asic |
| $m_{fmf}$ | **f**ree size-dependence | **m**onotonic | **f**ree size-dependence |
| $m_{ftf}$ | **f**ree size-dependence | **t**ime-dependent | **f**ree size-dependence |

[*]The letters in the subscript of the model name denote the growth, division, and loss parameterizations used in the model, respectively.

### Estimation of daily rates

We first assessed our models' ability to recreate the observed *Prochlorococcus* cell size distribution. Then, we examined whether an improved fit to the size distribution data resulted in improved model performance by comparing model estimates of daily average carbon fixation, carbon loss, and division rates to independent measurements from laboratory data. Finally, we investigated model estimated photosynthetic parameters.

As expected, the mean squared error (MSE) of the estimated cell size distribution decreased as the number of model parameters increased (Fig 4A). Critically, however, this improved fit did not correlate with better daily rate estimates. One of the most important parameters estimated by the models is the daily rate of cell division, see Eq (25). The observed daily division rate in the population was $0.63 \pm 0.01$ d$^{-1}$ (1 standard deviation interval). However, the simplest model $m_{bmx}$ overestimated this rate by nearly a factor of two (Fig 4B; $1.06 \pm 0.05$ d$^{-1}$). This may stem from the fact that this model did not include carbon loss; thus, it attributed any reduction in cell size to cell division. Model $m_{bmb}$, which adds respiratory/exudative carbon loss, was able to accurately estimate the daily division rate ($0.63 \pm 0.02$ d$^{-1}$), while all other models produced less accurate estimates, despite lower MSE of the estimated cell size distribution.

Model $m_{bmb}$ also performed well in estimating daily rates of carbon fixation and loss (Fig 4C and 4D). Again, the models with the best fit to the size distribution ($m_{fmf}$, $m_{ftf}$) exhibited

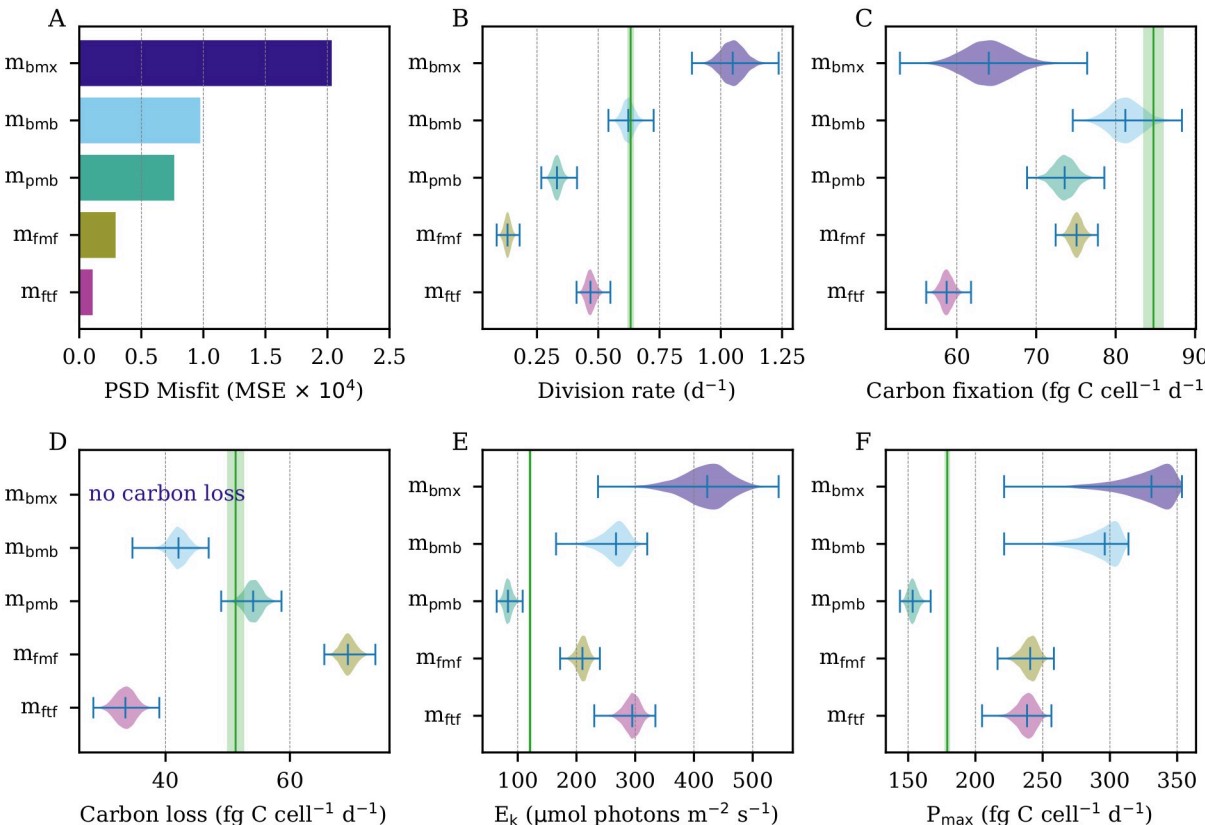

**Fig 4. Model estimated daily rate parameters.** (A) Mean squared error (MSE) of estimated proportions to the observed particle size distribution (PSD). (B) Estimated daily division rates. (C) Estimated daily carbon fixation. (D) Estimated daily carbon loss. (E) Estimated photosynthetic saturation parameter. (F) Estimated maximum photosynthetic rate. (B-F) Green vertical lines indicate ground truth calculated from data. Green shaded areas indicate uncertainty surrounding ground truth measurements. Model estimates shown as posterior distributions.

lower accuracy in their estimates of these rates. Interestingly, the addition of size-dependent carbon fixation ($m_{pmb}$) resulted in underestimation of daily carbon fixation (75.57 ± 1.00 fg C cell$^{-1}$ d$^{-1}$) and cell division (0.33 ± 0.02 d$^{-1}$) but improved estimates of daily carbon loss. The further addition of size-dependence in carbon loss ($m_{fmf}$) led to overestimates of daily carbon loss and even lower division rate estimates, indicating that this model attributes too much of the observed decreases in cell size to carbon loss rather than cell division. Model $m_{ftf}$, with added time-dependent division, underestimated daily rates of carbon fixation, division, and carbon loss.

Finally, we examined the photosynthetic saturation parameter $E_k$ and the maximum light-saturated photosynthetic rate $P_{max}$, two components of the mechanics of carbon fixation (see Carbon fixation section). Model $m_{bmx}$ shows the worst performance for these parameters, but $m_{bmb}$ also greatly overestimates both quantities despite accurate estimation of daily carbon fixation, highlighting potentially weak identifiability—i.e. similar daily carbon fixation rates can be obtained by different means, as carbon fixation decreases with higher values of $E_k$ but increases with higher values of $P_{max}$. These are examined further via simulation studies in the Supporting Information (S3 Text). Interestingly, $m_{pmb}$ had much more accurate estimates of the photosynthetic parameters, despite lower accuracy in overall daily carbon fixation. Size-dependent carbon loss ($m_{fmf}$) and time-dependent division ($m_{ftf}$) resulted in poorer estimates of the photosynthetic parameters relative to $m_{pmb}$.

Overall, the simplest model $m_{bmx}$ showed the poorest performance in estimation for nearly every category, highlighting the importance of accounting for carbon loss in our models. There is no model that performed best with respect to all the daily rate estimates we included in our tests; $m_{bmb}$ created the best division and carbon fixation estimates, while $m_{pmb}$ provided the best performance for $E_k$, $P_{max}$, and daily carbon loss.

## Estimation of hourly rates

In addition to the analysis of daily rate parameters, we examined the models' abilities to recreate population dynamics at hourly resolution (Fig 5) to determine whether discrepancies between model estimates and observations occur at a particular time of the diel cycle and to help us identify the relevant biological processes at play. While some of our models were able to estimate the daily rates of cell division, carbon fixation, and carbon loss accurately, the hourly patterns were more difficult to replicate (Fig 5A–5C). As expected by the relationship between cell size and hourly division rates (Fig 3), models that assume that cell division is only size-dependent ($m_{bmx}$, $m_{bmb}$, $m_{pmb}$, $m_{fmf}$) estimated the timing of cell division to be 4 to 8 hours too early (Fig 5A). On the other hand, model $m_{ftf}$, with both time-dependent division and size-dependent carbon fixation, was able to more accurately estimate the timing of cell division. However, this model underestimated division at dusk, thus leading to the inaccurate daily rates as discussed above. All models were able to capture the timing of carbon fixation, which is tied to the amount of incident light (Fig 5B). Yet, most models tended to underestimate the amount of fixed carbon, with $m_{bmb}$ coming closest to capturing the dynamics observed in the data. Surprisingly, the timing of carbon loss computed from the data (Fig 5C) closely matched that of carbon fixation. Our models tended to underestimate carbon loss during daytime peaks and overestimate it at night.

To further explore the estimated dynamics of division, carbon fixation, and carbon loss, we investigated the estimated proportions of cells undergoing each of these transitions as a function of cell size (Fig 5D–5F). The estimated shape of the size-division relationship tended to follow a sigmoidal pattern for all models: the fraction of dividing cells increases sharply above a critical size, which varied from 60 to 110 fg C depending on the model (Fig 5D). We note

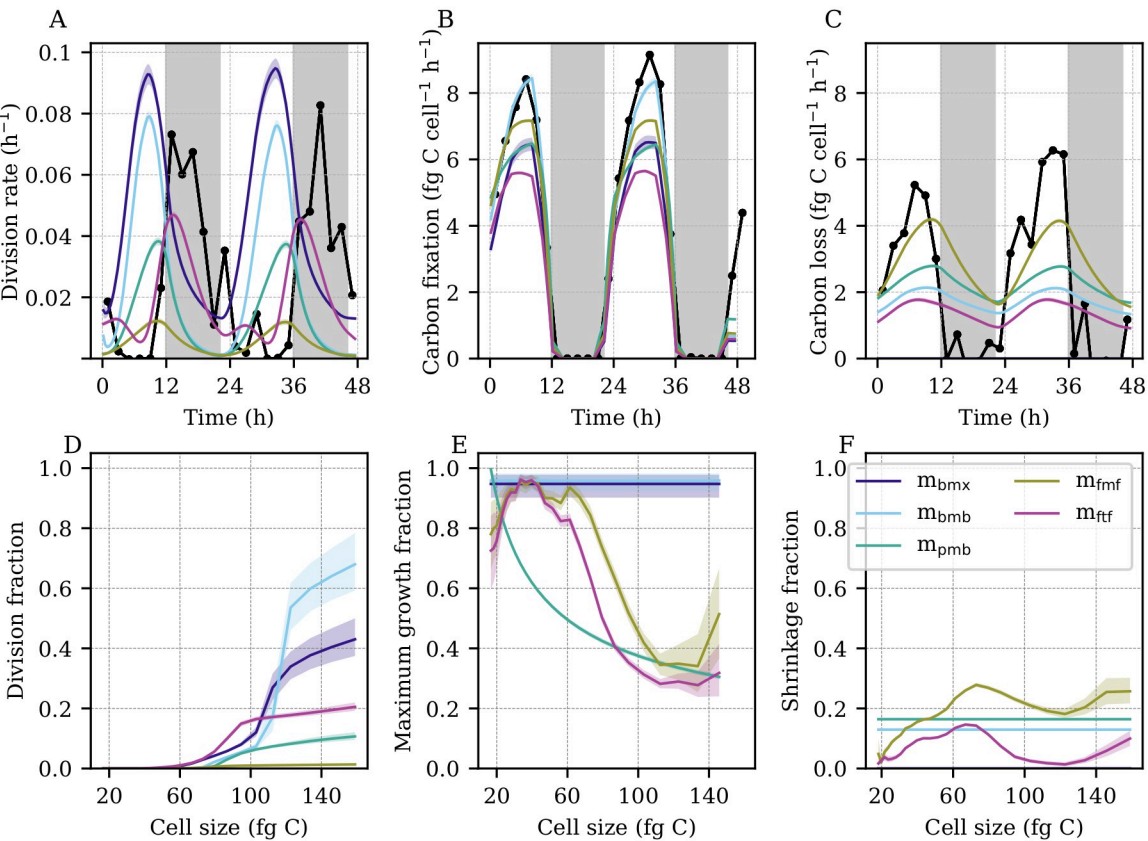

**Fig 5. Model estimated hourly rate parameters.** (A) Observed (black) and estimated (colored bands) hourly division rates. (B) Observed (black) and estimated (colored bands) hourly carbon fixation. (C) Observed (black) and estimated (colored bands) hourly carbon loss. (A-C) Black points indicate ground truth calculated from data. (D) Estimated cell division fraction as a function of cell size. (E) Estimated light-saturated cell growth (carbon fixation) fraction as a function of cell size. (F) Estimated cell shrinkage (carbon loss) fraction as a function of cell size. (A-F) Colored bands indicate model estimates. Shading indicates the first to third quartiles of the posterior distributions. (D-F) Fractions correspond to MPM transitions over a 20-minute time period.

that the model that best estimated the daily division rate ($m_{bmb}$) expected cell division to occur mostly in the largest size classes ($> 110$ fg C), which resulted in accurate amplitudes of hourly cell division rates, albeit at a 6-hour phase shift. In general, models that underestimated division ($m_{fmf}$, $m_{ftf}$) estimated smaller proportions of dividing cells within the larger size classes. However, $m_{bmx}$, which generally estimates a comparable or lower division fraction than $m_{bmb}$ at a given size, overestimates cell division. Because $m_{bmx}$ contains no carbon loss, it estimates more large cells to be present in the distribution, hence increasing the division rate relative to $m_{bmb}$ even if the division fraction is lower.

Meanwhile, model estimates of the size-dependence of carbon fixation generally estimated high values for the peak maximum growth fraction (Fig 5E). Models that assumed constant maximum growth ($m_{bmx}$, $m_{bmb}$) estimated this fraction to be near one. Interestingly, models with a free parameterization of size-dependent carbon fixation ($m_{fmf}$, $m_{ftf}$) generally estimated larger cells to have a lower maximum growth fraction, as in the power-law formulation ($m_{pmb}$). The estimated fractions of cell shrinkage tended to be significantly lower than the fractions of maximum growth, ranging from negligible to about one-fifth of the peak maximum growth fraction (Fig 5E and 5F). In the two models with size-dependent carbon loss rates ($m_{fmf}$, $m_{ftf}$), the estimated fraction of cell shrinkage generally increased with cell size. However,

both models estimated a sharp drop near the same critical sizes at which the division fraction sharply rose, suggesting that the models assign the decreases in cell size to cell division rather than carbon loss for larger but not smaller cells. These results suggest a trade-off of daily and hourly rate estimates between our models: models that produced some of the most accurate daily estimates of cell division, carbon fixation, and carbon loss showed a systematic offset in timing of cell division, while the models which accurately captured the timing often performed less well in estimating the daily average rate.

### Posterior parameter distributions

As the cell size distribution is used for model fitting, a model may be able to accurately capture the net effect of the parameters despite failing to accurately capture the value of each parameter individually, highlighting potential weak parameter identifiability. We therefore examined the bivariate joint posterior distributions of estimated rates of daily cell division, carbon fixation, and carbon loss (which are composites of many model parameters) as well as photosynthetic parameters to better understand the mechanics of the MPMs and the interdependencies of their parameters. We focused on two models: $m_{bmb}$, which had the best overall performance on daily rates of cell division, carbon fixation, and carbon loss but failed to predict the timing of cell division, and $m_{ftf}$, which was best able to predict the timing of cell division but failed to provide accurate daily rates (Fig 6). A strong correlation between daily carbon fixation and

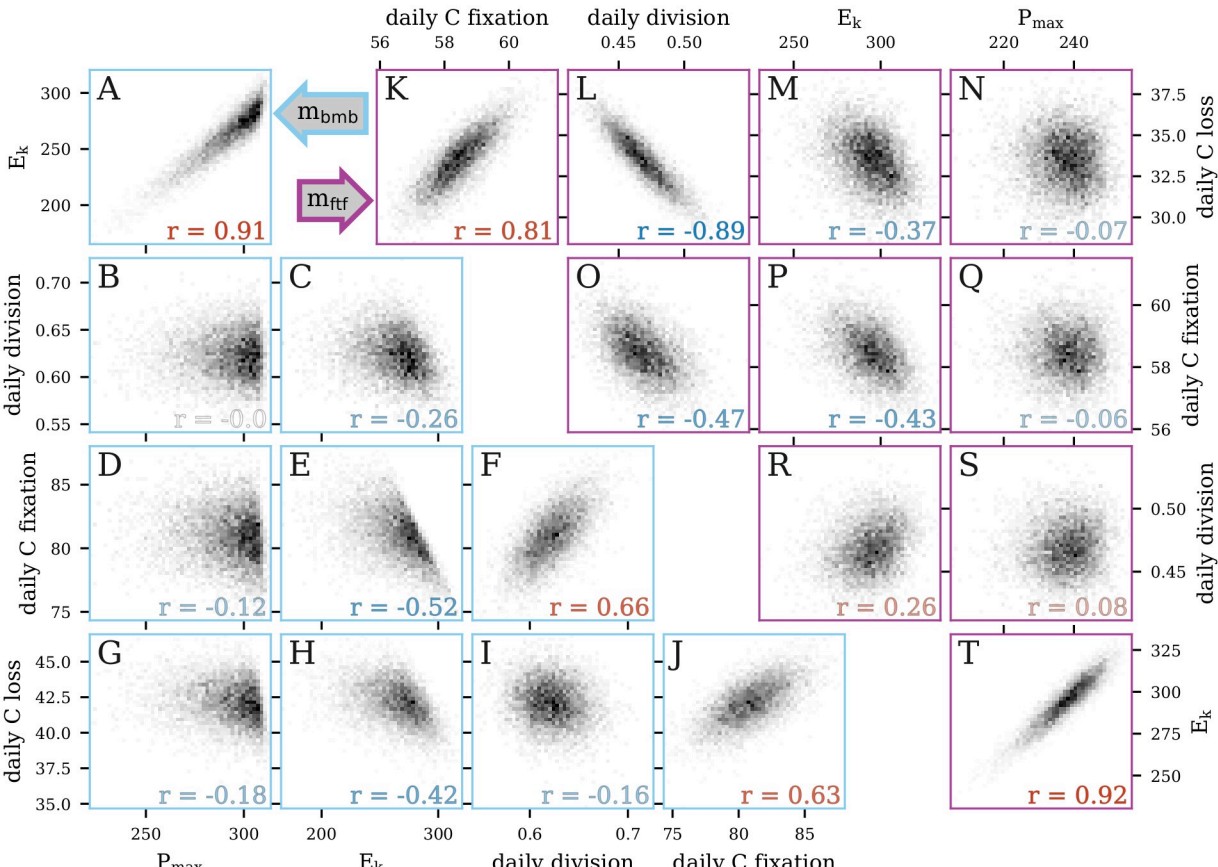

**Fig 6. Bivariate posterior distributions.** Scatter plots of the bivariate posterior distributions of select parameters for the models (A-J) $m_{bmb}$ and (K-T) $m_{ftf}$.

carbon loss was observed in the posterior distributions of both models (r = 0.63 and 0.81 for $m_{bmb}$ and $m_{ftf}$, respectively; Fig 6J and 6K), which was expected since the carbon fixed by photosynthesis fuels respiration and exudation. However, the relationship between carbon fixation and cell division differed between the two models (Fig 6F and 6O). Carbon fixation and cell division were positively correlated (r = 0.66) in $m_{bmb}$, which makes intuitive sense since the faster the cells grow, the faster they divide (Fig 6F), while a negative correlation (r = -0.47) was observed in $m_{ftf}$ (Fig 6O). This negative relationship likely stems from the fact that daily division rate and carbon loss in $m_{ftf}$ were strongly negatively correlated (r = -0.89, Fig 6L), while this relationship was much weaker in $m_{bmb}$ (r = -0.16, Fig 6I). As carbon fixation and carbon loss are tightly correlated, carbon loss may mediate the observed negative relationship between carbon fixation and daily division in $m_{ftf}$, making it more difficult for this model to disentangle these two processes than in $m_{bmb}$.

The shape of the posterior distribution highlights the strong relationship between $P_{max}$ and $E_k$ (Fig 6A and 6T); increases in $P_{max}$ and reduction of $E_k$ both increase carbon fixation in different ways (see Eq (7)), which would explain why $m_{bmb}$ could accurately estimate daily carbon fixation albeit with inaccurate estimates of photosynthetic parameters. The strong dependence structure between parameters shows that it is important to consider the joint distributions of the parameters and not solely focus on the marginal posterior distribution for each parameter. It also demonstrates that the size-distribution data itself cannot uniquely constrain all parameters, emphasizing the importance of prior knowledge and the prior distribution for limiting the parameter distributions.

## Observation sensitivity experiments

In order to quantify the impact of changes in the size distribution data on model parameter estimates, we performed two sets of experiments. In the first, we used a sliding window approach to assess the effect of changing the start time of the 48-hour time series on model output. In the second, we studied the robustness of the models to changes in the sampling resolution of observations.

In the sliding window experiment, we extended the normalized size distribution time series by appending the data to itself, thereby creating a four-day dataset. Then, we estimated parameters and initial conditions within a 48-hour window that was moved forward in time in four-hour increments. Details about the setup of the sliding window experiments and their results can be found in the Supporting Information (S1 Text). With the exception of $m_{bmx}$, all models exhibited a high degree of stability in their estimates for each window, indicating that the starting time of the model fitting procedure had a very limited effect on the models' parameter estimates. Some deviations were noticeable, however, when the window start time was near the peak of the cell size distribution, at which the difference between observations and model estimates is most pronounced. For $m_{bmx}$, estimates showed a high degree of variability among windows, suggesting that the results of this model may not be as stable or reliable as the others.

In the second set of experiments, we performed holdout validation experiments, in which time points of the size distribution data were withheld from the training data used for model fitting. This holdout data was then used as a testing set, and we computed the error for both datasets in order to examine our models' out-of-sample performance and the stability of the parameter estimation relative to the full dataset. We conducted three experiments, sequentially removing an increasing amount of equally spaced data, roughly mimicking lab experiments in which measurements were collected at lower resolution. This procedure ensured that the daily cycle was sampled well, and both days are represented equally in the training data. More details of this analysis can be found in the Supporting Information (S2 Text). We found that

parameter estimates and the observed cell size distribution remained stable when up to half of the data was removed from training, but out-of-sample performance deteriorated and parameter estimates differed significantly from those computed from the full data when two-thirds of the data was removed. This result suggests that our model could be applied to time series data collected at 4 hour intervals and still provide accurate estimated daily rates of cell division, carbon fixation, and carbon loss.

## Discussion

In this work, we extended size-structured MPMs to recover additional key biological processes that dictate phytoplankton cell growth, shrinkage, and division. Our investigation focused on a laboratory culture of the picocyanobacterium *Prochlorococcus*, whose dynamics over the diel cycle have been extensively studied [27]. We developed five models that differed in their parameterizations of changes in cell size. In addition to a size-dependent relationship for cell growth and time-dependence in cell division, we considered respiratory and exudative carbon loss in our models, which had previously been omitted in similar studies [18–24]. We implemented our models within a Bayesian framework, which permitted us to incorporate prior information into the analysis to regularize parameter inference and avoid biologically implausible parameter values.

The selection of priors is a requirement for the Bayesian inference procedure. When scientific knowledge is available to determine plausible parameter values, this can be formally incorporated into the inference and resulting estimates; otherwise, uninformative priors [31] can be used, which include broad uniform distributions or the so-called Jeffreys prior [32]. However, constraining complex models with uninformative priors may lead to poor identifiability and numerical instability. In this case, it may be useful to conduct additional studies to learn about plausible parameter ranges so that information can be brought into model fitting or use an approach known as Empirical Bayes, which aims to construct a prior distribution that is consistent with the data before formally fitting the model [33].

Herein, we showed that size-structured MPMs can be used to estimate not only rates of cell division but also carbon fluxes, offering the potential to connect microbial growth processes to the carbon cycle. The addition of carbon loss, which allows cells to shrink in size through a process other than cell division, improved the accuracy of model estimates and the fit to the size distribution data, with $m_{bmb}$ successfully recovering the measured daily rates of cell division, carbon fixation, and carbon loss (Fig 4B–4D). More complex models, such as those with size-dependent carbon fixation and time-dependent cell division, provided better fits to the cell size distribution and photosynthetic parameter estimates but showed worse model performance in recovering the observed daily rate parameter values. This result indicates that model fit to the observed cell size distribution cannot be used alone as a proxy for overall model performance.

As expected from the lack of correlation between cell size and hourly division rate in the laboratory data (Fig 3), most of our models consistently predicted the peak of cell division about 4–8 hours earlier than observed in the data (Fig 5 and S1 Fig). This offset stemmed from the assumption that cell division (i.e. the separation of a single cell into two daughter cells) occurs instantaneously once the cells reach a certain size. While this assumption may be reasonable on daily time scales, it becomes problematic at hourly resolution; cell division is a complex process involving many components, each highly regulated to ensure that the separation of the cell into two daughter cells occurs only once DNA synthesis is completed, which takes between 4 and 6 hours depending on the strain and culture conditions [27, 29]. Here, the peak of DNA synthesis coinciding with the peak of cell size [27] suggests that cell size dictates

the onset of DNA replication rather than the final separation of the cell into two daughter cells. Due to their greater flexibility, models with time-dependent division and size-dependent carbon fixation successfully captured the timing of cell division but failed to obtain accurate rate estimates. Interestingly, models with a free parameterization of size-dependent carbon fixation ($m_{fmf}$ and $m_{ftf}$) estimated less carbon fixation and more carbon loss in the large size classes which contains a large fraction of dividing cells (Fig 5E and 5F). This result suggests that dividing and non-dividing *Prochlorococcus* cells may have a different carbon metabolism, as observed in other organisms [34]. Ultimately, the choice of model will depend on the goal of the particular application. Our simpler models offered greater interpretability and accuracy of daily rate parameters, while more complex models were able to recover the timing of cell division at the cost of additional computation time and the requirement of stronger prior information.

Finally, we consider further potential future directions for this work. One of the interesting results in this study is the offset in the predicted and observed timing of division for the models with the most accurate daily division rate estimates. While the addition of time- and size-dependencies for cell division, carbon fixation, and loss allowed our more complex models to capture the timing of cell division, their estimates of the magnitude of division and other rate parameters suffered. As stated above, we hypothesize that carbon metabolism differs between dividing and non-dividing cells, yet our current modeling framework requires extension of the stage structure to encapsulate this information in order to test such a hypothesis. A hybrid age- and size-structured MPM may therefore be better suited to assess the importance of including cell division duration to more accurately simulate the timing of *Prochlorococcus* division, though this would expand the state-space of our models and require additional computation.

The flexibility of our modeling framework provides a valuable basis for examining and evaluating MPM results in the face of more complex datasets, which could further improve our understanding of the dynamics of marine microorganisms and their contributions to the carbon cycle. An exciting future extension of this work is application to *in-situ Prochlorococcus* and *Synechococcus* datasets obtained from shipboard flow cytometers [35]. Here, we tested our models on a highly synchronized population of *Prochlorococcus* grown under laboratory conditions, but we expect natural populations to be less synchronized and exhibit noisier dynamics over the diel cycle. Additional processes not accounted for in this study, such as grazing and viral lysis, which could potentially affect phytoplankton size distributions, will need to be tested. The application of our models to field data will be addressed in future work.

## Materials and methods

### Experimental data

A dataset of laboratory experiment time-series measurements of a high-light adapted strain of *Prochlorococcus* [27] collected during the exponential phase of batch growth over two simulated day-night cycles (Fig 2) was used to estimate biological parameters. We used changes in cell abundance over time to calculate division rates, since cell mortality is assumed to be negligible in exponentially growing cultures. A suite of measurements, which include cell size distributions and rates of carbon fixation, were collected at 2 hour intervals for a period of 50 hours to capture two complete diel cycles. Cell size distributions were inferred from flow-cytometry based forward-angle light scatter measurements (FALS). FALS signals normalized by calibration beads were converted to a proxy of mass using the relationships $M = FALS^{1/1.74}$ [36] and then converted to carbon quotas, assuming an average carbon quota of 53 fg C cell$^{-1}$ [27]. $^{14}$C-Photosynthetron experiments were conducted in duplicate at each time point to derive carbon fixation rates, maximum photosynthesis rates, and the photosynthetic saturation

parameter. Short (1 hour) incubation times were used to approximate gross carbon fixation rates. Using the 2-hourly cell abundance measurements ($a_t$), average cell size measurements ($s_t$) and approximate carbon fixation rates ($f_t$), we then estimated carbon loss rates ($l_t$) every 2 hours, using

$$s_{t+1} = s_t \; \frac{a_t}{a_{t+1}} + dt \; (f_t - l_t), \tag{2}$$

where $dt$ is the two hour time step between measurements. Measurements of photosynthetically active radiation (PAR) were collected every 2 hours. Note that of these measurements, only the cell size distribution and PAR data were used in model fitting. Estimates of division, carbon fixation, carbon loss, and photosynthetic parameters were used only to provide ground truth values and are not used in the model fitting procedure.

## Microbial matrix population models

The aim of the MPM applied to microbial populations is to mechanistically describe the evolution of the population over a day/night cycle. Typically, the target of inference is the daily division rate, which cannot be measured directly from changes in cell abundance measured in the field due to cell mortality caused by grazing and viral lysis as well as physical processes that can add or remove cells from the sampled population. Thus, in order to estimate this quantity, we infer it via observed changes in the relative abundance distribution over time. Past work has accomplished this by focusing on modeling two cellular processes: cell division and carbon fixation; in this work, we additionally consider carbon loss. We tested five MPMs involving these processes that varied in their complexity. All inference was carried out using the Bayesian modeling software Stan, see the Implementation section below.

**Preliminaries.** The MPM operates on discrete scales in both cell size and time. Therefore, there are two user-defined discretization parameters: $\Delta v \in \mathbb{R}^+$ is the size discretization parameter and $dt \in \mathbb{R}^+$ is the time discretization parameter in hours. We choose the former such that $(1/\Delta v) \in \mathbb{N}$ so that division corresponds to shifting $1/\Delta v$ size classes, see (3). We choose the latter to match our observation resolution; as the dataset has observations every 2 hours, we enforce $dt^{-1} \in \mathbb{N}$. In addition, we define $m \in \mathbb{N}$ the total number of discrete size classes and $v_1$ the minimum possible cell size, to define $m + 1$ size class boundaries:

$$v_i = v_1 \, 2^{(i-1)\Delta v} \; \forall \; i \in \{1, 2, \ldots, m+1\}. \tag{3}$$

If a cell is of size $x$ where $v_i \leq x < v_{i+1}$, then the cell belongs to size class $i$. Furthermore, we denote $j := 1/\Delta v + 1$ so that $v_j = 2v_1$, i.e. only cells of size class $j$ or greater can undergo cell division, see (14). For size-dependent parameterizations (see (7)), we treat cells in size class $i$ as having size

$$\bar{v}_i = \sqrt{v_i v_{i+1}}, \tag{4}$$

that is, they are assigned the geometric mean of the size class boundaries. In this work, we set $m = 27$, $\Delta v = 1/8$, $dt = 1/3$ hour, and $v_1 = 16$ fg C.

**Model inputs.** The observations $\{\boldsymbol{n}_k\}_{k=0}^{K-1}$ consist of cell counts across the $m$ discrete size classes at $K \in \mathbb{N}$ time points; that is, $\boldsymbol{n}_k \in \mathbb{N}^m \; \forall \; k \in \{0, 1, 2, \ldots, K-1\}$. We denote the set of observation times as $\mathcal{T} = \{t_0, t_1, \ldots, t_{K-1}\}$, where $t_k \in \mathbb{N}$ refers to the time in hours of the $k^{\text{th}}$ observation. For each $k$, we also define the simplex $\boldsymbol{w}_k = \boldsymbol{n}_k/N_k \in \Delta^{m-1}$, where $N_k = \sum_{i=1}^{m} n_k^{(i)}$ is the total number of cells observed at time $t_k$. Observations also include measurements of photosynthetically active radiation (PAR). This auxiliary data is linearly interpolated at the

times $\mathcal{T}^* := \{0, dt, 2dt, \ldots, T\}$, where the times are in hours; this information is used to estimate carbon fixation, which is assumed to be a function of PAR. We denote these values as $E := \{E(t)\}_{t \in \mathcal{T}^*}$ and treat them as fixed throughout the analysis. In our case, we have $T = 46$, $K = 24$, and $\mathcal{T} = \{0, 2, 4, \ldots, 46\}$.

**Parameterizations.** Our models aim to quantify the rates of three key biological processes: cell division, carbon fixation, and carbon loss. These rates are deterministic functions of the parameter vector $\theta$, which describes the dynamics of these processes, while the concentration parameter $\sigma$ allows for overdispersion in the data. We can divide the parameter vector $\theta$ into four components $\theta = (\theta_\delta, \theta_\gamma, \theta_\rho, \omega_0)$. The first three components correspond to each of the three cellular process we aim to model: cell division, carbon fixation, and carbon loss, respectively. The fourth defines the statistical mean of the cell size distribution at $t = 0$. The mean of the cell size distribution at each time point is a deterministic function of the model parameters $\theta$; we consider the data to be stochastically distributed around this mean; see Observation model for details. We use Stan's default prior for the initial condition simplex $\omega_0 \in \Delta^{m-1}$. We describe the parameterizations of the remaining three components in the following; see also Fig 7.

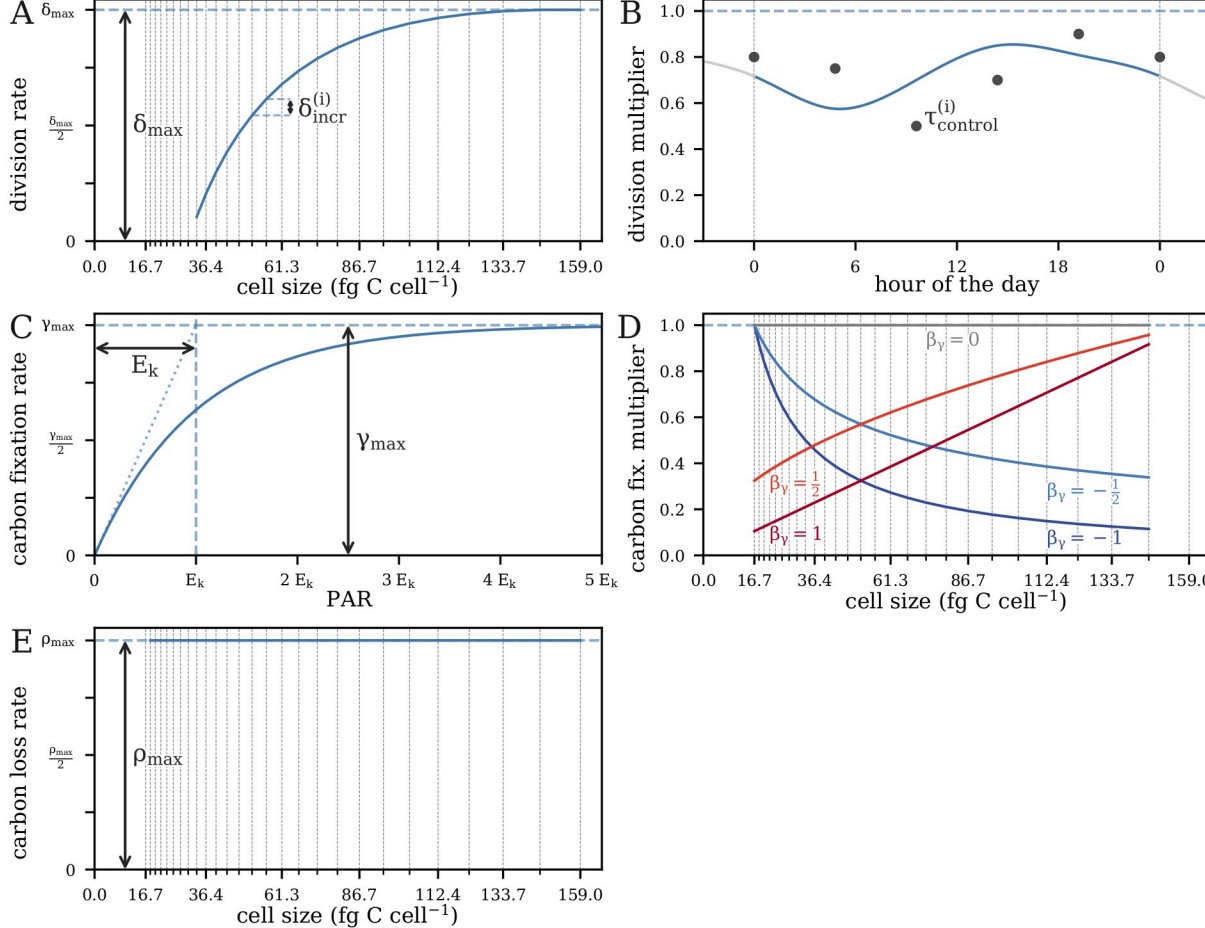

**Fig 7. Functional forms of model rate parameter dependencies on size, light, and time (compare Table 1).** (A) Examples of a monotonic relationship between size and division rate, and (B) a time-dependent relationship between the time of day and division rate. (C) The shape the light-dependent carbon fixation rate and examples of (D) power-law size-dependence of carbon fixations for 5 values of the parameter $\beta_\gamma$; the "basic" parameterization is identical to $\beta_\gamma = 0$. (E) The "basic" size-dependence of carbon loss. Vertical lines in panels with size-dependence denote the center of each size class. Examples of "free" carbon fixation or carbon loss parameterizations are not shown.

**Cell division.** The cell division proportions $\delta_i(t)$ are parameterized as

$$\delta_i(t) \quad = \begin{cases} 0 & i < j \\ \dfrac{dt}{24}\, \delta_{\max}\, q(t)\, \sum_{k=j}^{i} \delta_{\text{incr}}^{(k)} & i \geq j \end{cases} \tag{5}$$

where $\delta_{\max} \in [0, 24dt^{-1}]$ is the maximum division quotient, $q(t)$ is a function that induces time-dependence in division, and $\boldsymbol{\delta}_{\text{incr}} \in \Delta^{m-j}$ is a simplex that defines the relative increase in the division quotient for each size class. Recall that the division parameterization of each model is indicated by the second letter in its subscript (Table 1). For models with time-invariant division ($m_{\cdot m \cdot}$), $q(t) = 1$. The parameter $\delta_{\max}$ is normalized by $dt$ in units of days to better facilitate comparisons among models that vary in their values of $dt$; hence, $(dt/24)\delta_{\max} \in [0, 1]$. The parameter $\delta_{\text{incr}}$ allows us to constrain cell division to be monotone without imposing a specific functional form of the relationship between cell size and cell division. For models with time-dependent division ($m_{\cdot t \cdot}$), $q(t)$ is estimated using a periodic cubic spline with 6 knots and associated control points $\boldsymbol{\tau}_{\text{control}} := (\tau_{\text{control}}^{(1)}, \ldots, \tau_{\text{control}}^{(6)})^T \in \mathbb{R}^6$. Thus, we have

$$\boldsymbol{\theta}_{\delta} \quad = \begin{cases} (\delta_{\max}, \boldsymbol{\delta}_{\text{incr}}) & m_{\cdot m \cdot} \\ (\delta_{\max}, \boldsymbol{\delta}_{\text{incr}}, \boldsymbol{\tau}_{\text{control}}) & m_{\cdot t \cdot} \end{cases} \tag{6}$$

**Carbon fixation.** The cell growth proportions are parameterized as

$$\gamma_i(t) \quad = \begin{cases} \dfrac{dt}{24(2^{\Delta_v} - 1)}\, \gamma_{\max}\, s_{\gamma}^{(i)} \left(1 - \exp\left(\dfrac{-E(t)}{E_k}\right)\right) & i < m \\ 0 & i = m \end{cases} \tag{7}$$

where $\gamma_{\max} \in [0, 24(2^{\Delta_v} - 1)/dt]$ is the maximum cell growth quotient, $s_{\gamma}^{(i)}$ is a function that induces size-dependence in carbon fixation, and $E_k \in \mathbb{R}$ is a photosynthetic saturation parameter. Recall that $E(t)$ refers to the incident PAR at time $t$. The parameter $\gamma_{\max}$ is normalized by both the choices of time and size discretization to facilitate comparisons between models with different choices of discretization parameters. Recall also that the carbon fixation parameterization is indicated by the first letter in each model subscript (Table 1). For models without size-dependent carbon fixation ($m_{b\cdot\cdot}$), $s_{\gamma}^{(i)} = 1$. For models with a power-law carbon fixation ($m_{p\cdot\cdot}$),

$$s_{\gamma}^{(i)} \quad = \begin{cases} (\bar{v}_i/\bar{v}_m)^{\beta_{\gamma}} & \beta_{\gamma} \geq 0 \\ (\bar{v}_i/\bar{v}_1)^{\beta_{\gamma}} & \beta_{\gamma} < 0 \end{cases} \tag{8}$$

where $\beta_{\gamma} \in \mathbb{R}$ is a parameter that governs the power-law dependence of carbon fixation on size. For models with a free carbon fixation relationship ($m_{f\cdot\cdot}$), $s_{\gamma}^{(i)}$ is itself estimated as a parameter separately for each size class. Thus, we have

$$\boldsymbol{\theta}_{\gamma} \quad = \begin{cases} (\gamma_{\max}, E_k) & m_{b\cdot\cdot} \\ (\gamma_{\max}, E_k, \beta_{\gamma}) & m_{p\cdot\cdot} \\ (\gamma_{\max}, E_k, \boldsymbol{s}_{\gamma}) & m_{f\cdot\cdot} \end{cases} \tag{9}$$

For estimation of the light-saturated photosynthetic rate $\text{P}_{\max}$, we define the light-saturated

growth proportion

$$
\begin{aligned}
\gamma_i^\star(t) &= \lim_{E(t)\to\infty} \gamma_i(t) \\
&= \frac{dt}{24(2^{\Delta_v} - 1)} \gamma_{\max} s_\gamma^{(i)}.
\end{aligned}
\tag{10}
$$

Then, $P_{\max}$ is defined as the amount of carbon fixed when $\gamma_i(t)$ is replaced by $\gamma_i^\star(t)$ for all size classes $i$ and all time points $t \in \mathcal{T}^\star$.

**Carbon loss.** The carbon loss proportions are parameterized as

$$
\rho_i(t) = \begin{cases}
0 & i = 1 \\
\dfrac{dt}{24(2^{\Delta_v} - 1)} \rho_{\max} s_\rho^{(i)} & i > 1
\end{cases}
\tag{11}
$$

where $\rho_{\max} \in [0, 24(2^{\Delta_v} - 1)/dt]$ is the maximum cell shrinkage quotient normalized in the same way as $\gamma_{\max}$ and $s_\rho^{(i)}$ induces size-dependence in carbon loss. Recall that the carbon loss parameterization is indicated by the third letter in each model subscript (Table 1). For models with no respiration ($m_{\cdot\cdot x}$), $s_\rho^{(i)} = \rho_{\max} = 0$ and $\boldsymbol{\theta}_\rho$ is not included among the model parameters. For models with basic respiration ($m_{\cdot\cdot b}$), $s_\rho^{(i)} = 1$. For models with free size-dependent respiration ($m_{\cdot\cdot f}$), $s_\rho^{(i)}$ is itself estimated as a parameter as with $s_\gamma^{(i)}$. Thus, for models with respiration, we have

$$
\boldsymbol{\theta}_\rho = \begin{cases}
\rho_{\max} & m_{\cdot\cdot b} \\
(\rho_{\max}, \boldsymbol{s}_\rho) & m_{\cdot\cdot f}
\end{cases}
\tag{12}
$$

**Projection matrix.** The projection matrices are the core of a MPM: they project the population state forward in time. The population state at a given time point can be thought of as the statistical mean estimate of the *Prochlorococcus* cell size distribution. In this application, the projection matrices are deterministic functions of the parameters $\boldsymbol{\theta}$; the dependence on light $\boldsymbol{E}$ and the time-dependent division parameterization (for $m_{\text{fff}}$) add time-dependence. The projection matrices define the dynamics of the microbial population through the three cellular processes described above: cell division, carbon fixation, and carbon loss. We assume that any individual cell can only undergo one of these three processes in each $dt$ time step (it may also remain in the same size class). Thus, for each $k$, we first construct a set of matrices $\left\{ A_k^{(\ell)}(\boldsymbol{\theta}, \boldsymbol{E}) \right\}_{\ell=0}^{r_k - 1}$, where $r_k := (t_{k+1} - t_k)/dt$ is the number of $dt$ time steps between time $t_k$ and time $t_{k+1}$. Once these matrices are defined, we have for each $k$:

$$
\boldsymbol{B}_k(\boldsymbol{\theta}, \boldsymbol{E}) = \prod_{\ell=0}^{r_k - 1} A_k^{(r_k - 1 - \ell)}(\boldsymbol{\theta}, \boldsymbol{E}).
\tag{13}
$$

Each matrix $A_k^{(\ell)}(\boldsymbol{\theta})$ projects the population from time $t_k^{(\ell)} := t_k + \ell dt$ to time $t_k^{(\ell+1)} := t_k + (\ell + 1)dt$.

Let $\delta_i(t) \in [0, 1]$ denote the proportion of cells in size class $i$ that divide in one $dt$ time step at time $t$, $\rho_i \in [0, 1]$ the proportion of cells in size class $i$ that shrink one size class in one $dt$ time step given that they do not divide, and $\gamma_i(t) \in [0, 1]$ the proportion of cells in size class $i$ that grow one size class in one $dt$ time step at time $t$ given that they neither divide nor shrink. Then, recalling that $j$ denotes the index of the smallest size class which can undergo division,

the entries of each matrix $A_k^{(\ell)}(\boldsymbol{\theta})$ are defined as follows:

$$\textbf{division}: \qquad a_{k_{(i-j+1,i)}}^{(\ell)}(\boldsymbol{\theta}) = 2\,\delta_i(t_k^{(\ell)}) \qquad \text{for } j \le i \le m, \tag{14}$$

$$\textbf{growth}: \qquad a_{k_{(i+1,i)}}^{(\ell)}(\boldsymbol{\theta}, \boldsymbol{E}) = \begin{cases} \gamma_1(t_k^{(\ell)}) & \text{for } i = 1 \\[2mm] (1-\rho_i)\,\gamma_i(t_k^{(\ell)}) & \text{for } 2 \le i \le j-1 \\[2mm] (1-\delta_i(t_k^{(\ell)}))\,\gamma_i(t_k^{(\ell)})\,(1-\rho_i) & \text{for } j \le i \le m-1 \end{cases}, \tag{15}$$

$$\textbf{loss}: \qquad a_{k_{(i-1,i)}}^{(\ell)}(\boldsymbol{\theta}) = \begin{cases} \rho_i & \text{for } 2 \le i \le j-1 \\[2mm] (1-\delta_i(t_k^{(\ell)}))\,\rho_i & \text{for } j \le i \le m \end{cases}, \tag{16}$$

$$\textbf{stasis}: \qquad a_{k_{(i,i)}}^{(\ell)}(\boldsymbol{\theta}, \boldsymbol{E}) = \begin{cases} 1-\gamma_1(t_k^{(\ell)}) & \text{for } i = 1 \\[2mm] (1-\gamma_i(t_k^{(\ell)}))\,(1-\rho_i) & \text{for } 2 \le i \le j-1 \\[2mm] (1-\delta_i(t_k^{(\ell)}))\,(1-\gamma_i(t_k^{(\ell)}))\,(1-\rho_i) & \text{for } j \le i \le m-1 \\[2mm] (1-\delta_m(t_k^{(\ell)}))\,(1-\rho_m) & \text{for } i = m \end{cases}, \tag{17}$$

where again $t_k^{(\ell)} := t_k + \ell dt$. Here, only cell growth and stasis involve the PAR measurements $\boldsymbol{E}$. The coefficient 2 in Eq (14) reflects the fact that when a cell divides, it creates two daughter cells. This is the reason the normalization step (18) is needed to maintain the sum-to-one constraint and also the reason (26), which omits the normalization, is able to estimate the rate of cell division.

MPMs for microbial populations make projections differently from the formulation in (1). The counts are normalized at each time step so that we model the mean relative abundance:

$$\boldsymbol{\omega}_{k+1}(\boldsymbol{\theta}, \boldsymbol{E}) = \frac{B_k(\boldsymbol{\theta}, \boldsymbol{E})\boldsymbol{\omega}_k(\boldsymbol{\theta}, \boldsymbol{E})}{\sum_{i=1}^{m}\sum_{j=1}^{m} B_k^{(i,j)}(\boldsymbol{\theta}, \boldsymbol{E})\omega_k^{(j)}(\boldsymbol{\theta}, \boldsymbol{E})}. \tag{18}$$

Note that $\boldsymbol{\omega}_k(\boldsymbol{\theta}, \boldsymbol{E})$ is a deterministic function of the model parameters $\boldsymbol{\theta}$ and the interpolated PAR $\boldsymbol{E}$. Thus, we do not use the counts to estimate division rate directly, allowing for valid estimates even when mortality and physical movement of cells occur, so long as these processes do not affect the relative size distribution. We estimate the posterior distributions of the model parameters from their prior distributions and the likelihood of the data $\{\boldsymbol{n}_k\}_{k=0}^{K-1}$ given the parameters (see Observation model section).

**Observation model.** The observation model links the population state defined by the projection matrices (13) to the observed state. Thus, this model accounts for any deviations of the observations from the population states. The observations are assumed to arise from the

following statistical model:

$$n_k | \eta_k, \sigma, \theta \sim \text{Multinomial}(N_k; \eta_k) \tag{19}$$

$$\eta_k | \sigma, \theta \sim \text{Dirichlet}(\sigma \omega_k(\theta, E)) \tag{20}$$

$$\sigma \sim \pi_\sigma \tag{21}$$

$$\theta \sim \pi_\theta \tag{22}$$

where $\sigma$ is a real-valued concentration parameter, $\theta$ is a parameter vector, and $\pi$. denotes the corresponding prior distributions (see Table 2). Thus, similar to [19], the model likelihood can be written as

$$p\big(\{n_k\}_{k=0}^{K-1} | \theta, \sigma\big) = \prod_{k=0}^{K-1} \left\{ \frac{\Gamma(\sigma)N_k!}{\Gamma(N_k + \sigma)} \times \prod_{i=1}^{m} \left[ \frac{\Gamma\left(n_k^{(i)} + \sigma\omega_k^{(i)}(\theta, E)\right)}{\Gamma\left(\sigma\omega_k^{(i)}(\theta, E)\right)n_k^{(i)}!} \right] \right\}, \tag{23}$$

where $n_k^{(i)} \in \mathbb{R}$ is the $i^{\text{th}}$ entry of $n_k$ and $\omega_k^{(i)}(\theta, E) \in \mathbb{R}$ is the $i^{\text{th}}$ entry of $\omega_k(\theta, E)$, the population state for the $k^{\text{th}}$ observation. The posterior is proportional to the product of the likelihood and the prior distribution according to Bayes' theorem; thus, we have

$$p\big(\theta, \sigma | \{n_k\}_{k=0}^{K-1}\big) \propto p\big(\{n_k\}_{k=0}^{K-1} | \theta, \sigma\big) \pi(\theta, \sigma), \tag{24}$$

where the proportionality holds because the evidence
$p\big(\{n_k\}_{k=0}^{K-1}\big) = \int \int_{\theta,\sigma} p\big(\{n_k\}_{k=0}^{K-1} | \theta, \sigma\big) \pi(\theta, \sigma) d\theta d\sigma$ is constant with respect to the model parameters $(\theta, \sigma)$ [37].

**Table 2. List of model parameters.**

| Name | Used in | Description | Units | Bounds | Prior |
|---|---|---|---|---|---|
| $\omega_0$ | all models | initial conditions | – | simplex | Stan default |
| $\sigma$ | all models | concentration parameter | – | $[0, \infty)$ | Stan default |
| $E_k$ | all models | light-dependent growth parameter | $\mu$mol photons m$^{-2}$ s$^{-1}$ | $[0, 5000]$ | Normal(1000, 1000) |
| $\delta_{\max}$ | all models | maximum division rate | d$^{-1}$ | $\left[0, \frac{1}{\Delta_t}\right]$ | Stan default |
| $\delta_{\text{incr}}^{(i)}$ | all models | increment in division rate, size class $i$ | – | $[0, 1]$ | Stan default |
| $\gamma_{\max}$ | all but $m_{\text{f}\cdot\cdot}$ | maximum carbon fixation rate | d$^{-1}$ | $\left[0, \frac{1}{\Delta_{t^*}}\right]$ | Normal(10.0, 10.0) |
| $\beta_\gamma$ | $m_{\text{p}\cdot\cdot}$ | exponent in carbon fixation power law | – | $[-10, 10]$ | Normal(0, 0.1) |
| $\gamma_{\max}^{(i)}$ | $m_{\text{f}\cdot\cdot}$ | maximum carbon fixation rate, size class $i$ | d$^{-1}$ | $\left[0, \frac{1}{\Delta_{t^*}}\right]$ | Normal($\mu_\gamma, \sigma_\gamma$) |
| $\mu_\gamma$ | $m_{\text{f}\cdot\cdot}$ | hierarchical prior for mean of $\gamma_{\max}^{(i)}$ | d$^{-1}$ | $\left[0, \frac{1}{\Delta_{t^*}}\right]$ | Normal(10.0, 10.0) |
| $\sigma_\gamma$ | $m_{\text{f}\cdot\cdot}$ | hierarchical prior for s.d. of $\gamma_{\max}^{(i)}$ | d$^{-1}$ | $[0, \infty)$ | Exponential(0.1) |
| $\rho_{\max}$ | all but $m_{\cdot\cdot\text{f}}$ | maximum carbon loss rate | d$^{-1}$ | $\left[0, \frac{1}{\Delta_{t^*}}\right]$ | Normal(3.0, 10.0) |
| $\rho_{\max}^{(i)}$ | $m_{\cdot\cdot\text{f}}$ | maximum carbon loss rate, size class $i$ | d$^{-1}$ | $\left[0, \frac{1}{\Delta_{t^*}}\right]$ | Normal($\mu_\gamma, \sigma_\gamma$) |
| $\mu_\rho$ | $m_{\cdot\cdot\text{f}}$ | hierarchical prior for mean of $\rho_{\max}^{(i)}$ | d$^{-1}$ | $\left[0, \frac{1}{\Delta_{t^*}}\right]$ | Normal(10.0, 10.0) |
| $\sigma_\rho$ | $m_{\cdot\cdot\text{f}}$ | hierarchical prior for s.d. of $\rho_{\max}^{(i)}$ | d$^{-1}$ | $[0, \infty)$ | Exponential(0.1) |
| $\tau_{\text{control}}^{(i)}$ | $m_{\cdot\text{t}\cdot}$ | control point $i$ for time-dep. division spline | – | $[0, 1]$ | Beta(9, 1) |

**Model output.**    The primary goal of inference is the daily division rate $\mu$, defined as the exponential growth constant:

$$\mu = \frac{24}{T} \log \left( \frac{N_{K-1}}{N_0} \right). \tag{25}$$

Recall that $T = t_{K-1}$ is the time of the last observation in hours; thus, $T/24$ is the length of the time series in days. Because populations in their natural environments undergo cell loss due to cell mortality (due to grazing and viral lysis) and physical processes that can add or remove cells, a normalization step (18) was applied to estimate division rate based on relative cell abundance, as in past applications [18, 19, 21]. By removing the normalization step, we estimate the relative increase in cell number caused by cell division. Given parameter estimates $\hat{\boldsymbol{\theta}}$, projection matrix estimates $\hat{\boldsymbol{B}}_k(\hat{\boldsymbol{\theta}}, \boldsymbol{E})$, and initial state estimate $\hat{\boldsymbol{\omega}}_0$, we obtain the following estimator of the daily division rate:

$$\hat{\mu}(\hat{\boldsymbol{\theta}}, \boldsymbol{E}) = \frac{24}{T} \log \left( \sum_{i=1}^{m} \left\{ \left[ \prod_{k=0}^{K-1} \hat{\boldsymbol{B}}_k(\hat{\boldsymbol{\theta}}, \boldsymbol{E}) \right] \hat{\boldsymbol{\omega}}_0(\hat{\boldsymbol{\theta}}, \boldsymbol{E}) \right\} \right). \tag{26}$$

## Implementation

Parameter inference was carried out in the software package Stan [25]. This software performs Bayesian inference, where the target is the *posterior* distribution of the parameters, which reflects the probable values of these parameters given the model, our prior beliefs, and the data [38]. In order to generate samples from the posterior distribution, Stan implements a variant of the Hamiltonian Monte Carlo (HMC) algorithm [39, 40] which has been shown to have superior speed and performance for fitting complex, high-dimensional population dynamics models relative to other Markov Chain Monte Carlo (MCMC) methods for sampling from the posterior [41]. In particular, we use Stan's implementation of the No-U-Turn Sampler (NUTS) [42] to avoid manual selection of application-specific tuning parameters. We initially tried using Stan's implementation of variational inference, which, while faster, creates approximate results and provided less stable estimates than HMC using NUTS in our experiments. Thus, in all of the experiments presented here, we used HMC, which generated stable results in this study and generally provides asymptotic consistency [40]. The implementation of HMC in Stan uses automatic differentiation to provide the gradients needed to integrate Hamiltonian dynamics. The reader is directed to [43] for additional details on HMC in Stan.

Six HMC chains were run for 2000 MCMC iterations for each model. In accordance with the Stan default settings, the first 1000 samples of each of these chains were discarded as a warm-up period for the sampler to reach its stationary distribution. The $\hat{R}$ convergence diagnostic [44] was monitored for all model fits to ensure $\hat{R} < 1.05$; otherwise, the sampling procedure was considered divergent. A comparison of the prior distributions of $m_{\mathrm{bmb}}$ and $m_{\mathrm{ftf}}$ with their corresponding posteriors can be found in the Supporting Information (S3 Fig).

In order to benchmark our results, we used Stan's optimization to compute the maximum likelihood estimator (MLE) for $m_{\mathrm{bmb}}$ and $m_{\mathrm{ftf}}$. However, the results were unstable and sensitive to initialization. To investigate the sensitivity of our inference to the prior distributions, we implemented $m_{\mathrm{bmb}}$ and $m_{\mathrm{ftf}}$ with flat priors, so that the posterior distribution is proportional to the likelihood. For $m_{\mathrm{bmb}}$, this gave virtually identical results. For $m_{\mathrm{ftf}}$, the model failed to converge, indicating that stronger prior information is necessary to remove potential identifiability issues introduced by the additional parameters for size- and time-dependent processes.

All code used to process the data, fit models, and produce visualizations is publicly available on GitHub [45].

## Prior distributions

The prior distributions are shown in Table 2. Priors were defined for the model parameters $\theta$ and $\sigma$, and not on rates of division, carbon fixation, and loss directly. Maximum cell division, carbon fixation and loss along with photosynthetic parameter values were chosen within biologically feasible ranges using information derived from literature [27, 46], otherwise the Stan default priors were used, corresponding to uniform priors [25].

## Supporting information

**S1 Text. Sliding window experiments.** Describes stability of model estimates across 48-hour sliding windows.
(PDF)

**S2 Text. Hold-out validation.** Describes model results when removing data points from the training set and examines model performance in predicting the held-out data.
(PDF)

**S3 Text. Estimating parameters from synthetic data.** Results of generating synthetic data from $m_{bmb}$ and $m_{ftf}$ and fitting the corresponding model to recover the underlying parameters.
(PDF)

**S1 Table. Expanded set of nine models.**
(PDF)

**S1 Fig. Daily rate estimates for all nine models.** (A) Mean squared error (MSE) of estimated proportions to the observed particle size distribution (PSD). (B) Estimated daily division rates. (C) Estimated daily carbon fixation. (D) Estimated daily carbon loss. (E) Estimated photosynthetic saturation parameter. (F) Estimated maximum photosynthetic rate. (B-F) Green vertical lines indicate ground truth calculated from data. Green shaded areas indicate uncertainty surrounding ground truth measurements. Model estimates shown as posterior distributions.
(TIF)

**S2 Fig. Hourly rate estimates for all nine models.** (A) Observed (black) and estimated (colored bands) hourly division rates. (B) Observed (black) and estimated (colored bands) hourly carbon fixation. (C) Observed (black) and estimated (colored bands) hourly carbon loss. (A-C) Black points indicate ground truth calculated from data. (D) Estimated cell division fraction as a function of cell size. (E) Estimated light-saturated cell growth (carbon fixation) fraction as a function of cell size. (F) Estimated cell shrinkage (carbon loss) fraction as a function of cell size. (A-F) Colored bands indicate model estimates. Shading indicates the first to third quartiles of the posterior distributions. (D-F) Fractions correspond to MPM transitions over a 20-minute time period.
(TIF)

**S3 Fig. Comparison of the prior and posterior distribution for biological parameters.** Prior pdf and histogram of posterior samples for select biological parameters of models $m_{bmb}$ (A-D) and $m_{ftf}$ (E-J). The Kullback–Leibler divergence of the marginal distribution ($D_{KL}$) quantifies the difference between prior and posterior distribution for each parameter.
(TIF)

**S4 Fig. HMC sampler chains for $m_{bmb}$.** Each of six parallel post-warmup sampling chains for four parameters of $m_{bmb}$. Points indicate individual samples and solid lines represent Gaussian smoothers. Each color corresponds to one of the six chains.
(TIF)

**S5 Fig. HMC sampler chains for $m_{ftf}$.** Each of six parallel post-warmup sampling chains for four parameters of $m_{ftf}$. Points indicate individual samples and solid lines represent Gaussian smoothers. Each color corresponds to one of the six chains.
(TIF)

## Acknowledgments

We would like to thank Zachary Johnson for sharing the data. We also thank Jacob Bien, Christopher A. Edwards and Mick Follows for their support of S.H., J.P.M, and Z.W., respectively. This work was initiated at the Simons Foundation Collaboration on Computational Biogeochemical Modeling of Marine Ecosystems (cbiomes.org) workshop on Bayesian analysis in marine ecosystems. We thank Helen Hill for workshop organization.

## Author Contributions

**Conceptualization:** Jann Paul Mattern, Kristof Glauninger, Gregory L. Britten, John R. Casey, Sangwon Hyun, Zhen Wu, E. Virginia Armbrust, Zaid Harchaoui, François Ribalet.

**Data curation:** Jann Paul Mattern, Kristof Glauninger, John R. Casey, François Ribalet.

**Formal analysis:** Jann Paul Mattern, Kristof Glauninger, François Ribalet.

**Funding acquisition:** E. Virginia Armbrust, Zaid Harchaoui, François Ribalet.

**Investigation:** Jann Paul Mattern, Kristof Glauninger, John R. Casey, François Ribalet.

**Methodology:** Jann Paul Mattern, Kristof Glauninger, Gregory L. Britten, John R. Casey, Sangwon Hyun, Zhen Wu, François Ribalet.

**Project administration:** Jann Paul Mattern, Kristof Glauninger, Gregory L. Britten, E. Virginia Armbrust, Zaid Harchaoui, François Ribalet.

**Resources:** Jann Paul Mattern, Kristof Glauninger, E. Virginia Armbrust, Zaid Harchaoui, François Ribalet.

**Software:** Jann Paul Mattern, Kristof Glauninger, François Ribalet.

**Supervision:** E. Virginia Armbrust, Zaid Harchaoui, François Ribalet.

**Validation:** Jann Paul Mattern, Kristof Glauninger, Gregory L. Britten, François Ribalet.

**Visualization:** Jann Paul Mattern, Kristof Glauninger, François Ribalet.

**Writing – original draft:** Jann Paul Mattern, Kristof Glauninger.

**Writing – review & editing:** Jann Paul Mattern, Kristof Glauninger, Gregory L. Britten, John R. Casey, Sangwon Hyun, Zhen Wu, E. Virginia Armbrust, Zaid Harchaoui, François Ribalet.

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
