## [Decision Letter · Decision Letter 0]

23 Sep 2021

Dear Mr. Glauninger,

Thank you very much for submitting your manuscript "A flexible Bayesian approach to estimating size-structured matrix population models" for consideration at PLOS Computational Biology.

As with all papers reviewed by the journal, your manuscript was reviewed by members of the editorial board and by several independent reviewers. In light of the reviews (below this email), we would like to invite the resubmission of a significantly-revised version that takes into account the reviewers' comments.

We cannot make any decision about publication until we have seen the revised manuscript and your response to the reviewers' comments. Your revised manuscript is also likely to be sent to reviewers for further evaluation.

Sincerely,

Inna Lavrik

Associate Editor

PLOS Computational Biology

Jason Haugh

Deputy Editor

PLOS Computational Biology

Reviewer's Responses to Questions

**Comments to the Authors:**

Reviewer #1: This paper explores and investigates several different formulations of a size-structured matrix population model to accurately estimate growth rate from hourly cell size distributions. Different model formulations are checked against laboratory data of the marine cyanobacterium Prochlorococcus to infer how well each model version estimates growth rate and other physiological relevant parameters.

The expansion of model formulations presented here is a valuable and worthwhile effort; if it is possible to capture additional physiological parameters of a phytoplankton population from cell-size distributions, the data would be quite informative. The authors attempt to expand and improve upon earlier models whose underlying formulas for transition matrices were not able to be linked directly to the cellular parameters of interest here (carbon fixation, carbon loss, and cell division).

However, identifiability issues raises serious concerns about whether model formulations and Bayesian approach are able to obtain valid inference:

1) A Bayesian approach requires that the evidence, P({n}), is constant in order to use the proportionality listed after equation 6, line 418. How can you ensure this partition function is constant?

2) Instability was encountered when attempting variational inference, which may indicate that the procedure is not producing valid estimates of the posterior using the likelihood and prior models. Authors do not provide posterior distributions for all parameters (only two are shown in Figure 6) nor how these estimates correspond to assumed priors, (e.g. for parameters E_k and rho_max given in Table 2); the presentation is not clear on whether posteriors are reasonable with respect to expert-choice priors. The presentation can be better if the reader is more directly convinced that the likelihood with chosen priors do in fact result in STAN sampling a realistic (rather than just an algebraically emergent) posterior. Perhaps the authors could include both the prior and posterior distributions (on the same plot) of a sensitive or difficult to constrain parameter from Table 2.

To this end, please show:

-Maximum likelihood estimates of parameters to show that priors and posterior estimates are reasonable; MLE should give similar values to mean posterior.

-MC sampler chains (i.e. as Reference 38, Figure 4) for the most difficult to constrain parameter to affirm ‘burn-in’, feasible models, and successful exploration of posterior.

- Plots of posterior probability for all parameters.

3) Please more fully discuss risk of over-parameterization. The paper itself ends with the admission that size-distribution data cannot constrain all the parameters. It is entirely possible that no model is able to accurately partition cell size changes between division, cell growth and carbon loss, so it is unclear how the authors are accounting for this and refining their model structures. It is also unclear why so many models are presented when some are very likely over-parameterized, especially the free-size dependence versions. The paper would benefit from a streamlining and selection of models, with over-parameterized models noted in the SI.

Please also discuss benefits and risks of incorporating more concrete cell biology within the model. For example, a time delay on cell division or no carbon loss during night would likely provide more simpler model formulations.

In addition, I have the following specific comments:

Title: The title as written is misleading; ‘flexible’, as written, addresses the Bayesian approach utilized, which appears to be standard. The title also suggests that the paper is about inference techniques for size-structure matrix population models (a very large class of models!), which is not the case. The flexibility that the authors are mentioning is presumably for model construction (not the inference procedure). The models implemented here are also highly specific to high-resolution, time series flow cytometry data of phytoplankton. The title should reflect this; please revise.

Line 23-24: Awkward/misplaced sentence; in-situ growth rate estimates are not really obscured by heterotrophic biomass nor detritus; it is because these rates cannot be obtained from abundance or carbon estimates alone (which are a composite of growth and loss). Please clarify.

Line 58-61: Authors are incorrect with regards to how previous models were evaluated. While goodness of fit is checked, model formulations in some papers were benchmakred against laboratory division rate data. Please revise. A more objective presentation of earlier studies is also needed across the manuscript; authors of previous model papers are presumably well-aware of their model limitations, and physiological measurements weren’t often the goal in these past studies. Risk of over-parameterization is only one of many reasons why other versions were not explored. Please do not assume intent unless explicitly stated in these earlier manuscripts.

Line 91: Ending sentence here is not entirely correct; there is no connection to the larger marine carbon cycle in this paper. The rate parameters perhaps may offer this, but the authors do not extrapolate to any larger cycles in this manuscript.

Lines 100-104: I’m not sure I entirely follow the arguments and the emphasis placed on cell size in relationship to hourly division rate. It is correct to compare plots of hourly division rate to mean cell size? Division produces an increase of small cells, such that a plot between mean cell size and division rate would not necessarily show a correlation (no large cells are expected with higher division rates, and after division, which must have happened to get a rate, cell size is no longer correlated to the process).

Furthermore, as the authors have count data in each size class, could the authors not present an analysis of how cell sizes shift in comparison to hourly division rate? For example, do the largest cells decrease in abundance immediately after dusk (as suggested by Fig.2), whereas medium cells decrease in abundance more towards the middle of the night? And is this all accompanied by corresponding increases in smaller cells? This would probably yield better insight into the timing of division and guidance for model division formulas. I agree that the process of cell division is not likely instantaneous and complete cell fission will likely take a few hours.

Discussion: The paper examines performance of each model version, but at the end, the reader is left wondering what perhaps is the best formulation going forward or where exactly the work should be going. Could the authors add additional recommendations or concrete decisions to guide readers?

Methods: Please provide example functional curves for cell growth, cell division and cell loss; these are useful for reader visualization and how different parameters affect each function.

Methods: Perhaps I missed it, but explicit code to call Stan with model formulations does not appear available.

Reviewer #2: The work presented in this manuscript represents an important advance in the use of demographic models for understanding the ecology of marine phytoplankton. It should eventually be published. There are, however, a few issues that need to be addressed in the current version of the manuscript.

1. One contribution of this paper is the application of Bayesian methods for parameter estimation to a size-structured matrix population model for marine phytoplankton. The application of Bayesian methods for matrix models is not particularly new. A second is the extension of the model to allow for the shrinking of cells (as a result of respiration). This second part is new. I would, the introduction of shrinking into the model introduces new parameters. I would like to see some evidence that these parameters can be estimated accurately from simulated data (where the true values are known). My intuition is that there are many combinations of the parameters that would produce the same sequence of size distribution, and, as a result identification may be an issue. The authors touch on this in the section that begins on line 258.

2. There is no free lunch: one must pay for the Bayesian approach by the specification of prior distributions on the parameters. In my opinion the authors do not discuss this cost sufficiently in the discussion. What does one do when there is no "prior knowledge?" How sensitive are the posterior distributions to the priors? This second question is important, and could (should?) be addressed with the simulation studies, but should also be addressed in estimation of the parameters for the laboratory data.

3. In a number of places in the manuscript (eg., lines 58-61, 346-348) where the authors claim that previous work was flawed because that previous work used a measure of goodness of fit to observed size distributions "as a proxy for overall model performance". At least for references [18], [19], [23] and [24] this was not the case. Instead, they judged model performance by how well the model could estimate division rate---the object of inference---compared with a "gold standard" method (dilution experiments) by calculating concordance.

**Have the authors made all data and (if applicable) computational code underlying the findings in their manuscript fully available?**

Reviewer #1: **No: **No code is provided that must have been used to construct model formulations and call Stan libraries

Reviewer #2: Yes

PLOS authors have the option to publish the peer review history of their article (what does this mean?). If published, this will include your full peer review and any attached files.

Reviewer #1: No

Reviewer #2: No
---

## [Editor Report · Decision Letter 1]

8 Dec 2021

Dear Mr. Glauninger,

We are pleased to inform you that your manuscript 'A Bayesian approach to modeling phytoplankton populationdynamics from size distribution time series' has been provisionally accepted for publication in PLOS Computational Biology.

Best regards,

Inna Lavrik

Associate Editor

PLOS Computational Biology

Jason Haugh

Deputy Editor

PLOS Computational Biology

---

## [Editor Report · Acceptance letter]

12 Jan 2022

PCOMPBIOL-D-21-00986R1 

A Bayesian approach to modeling phytoplankton populationdynamics from size distribution time series

Dear Dr Ribalet,

I am pleased to inform you that your manuscript has been formally accepted for publication in PLOS Computational Biology. Your manuscript is now with our production department and you will be notified of the publication date in due course.

With kind regards,

Agnes Pap
